# The Modulatory Effects of DMF on Microglia in Aged Mice Are Sex-Specific

**DOI:** 10.3390/cells11040729

**Published:** 2022-02-18

**Authors:** Virginia Mela, Aline Sayd Gaban, Eoin O’Neill, Sibylle Bechet, Aífe Walsh, Marina A. Lynch

**Affiliations:** 1Department of Medicine and Dermatology, Faculty of Medicine, University of Malaga, 29010 Malaga, Spain; virginiamelarivas@gmail.com; 2Trinity College Institute of Neuroscience, Trinity College Dublin, D02 DK07 Dublin, Ireland; sayd.aline@gmail.com (A.S.G.); oneille9@tcd.ie (E.O.); bechets@tcd.ie (S.B.); walsha69@tcd.ie (A.W.)

**Keywords:** sexual dimorphism, age, microglia, inflammation, metabolism, dimethyl fumarate

## Abstract

There is a striking sex-related difference in the prevalence of many neurodegenerative diseases, highlighting the need to consider whether treatments may exert sex-specific effects. A change in microglial activation state is a common feature of several neurodegenerative diseases and is considered to be a key factor in driving the inflammation that characterizes these conditions. Among the changes that have been described is a switch in microglial metabolism towards glycolysis which is associated with production of inflammatory mediators and reduced function. Marked sex-related differences in microglial number, phenotype and function have been described in late embryonic and early postnatal life in rodents and some reports suggest that sexual dimorphism extends into adulthood and age and, in models of Alzheimer’s disease, the changes are more profound in microglia from female, compared with male, mice. Dimethyl fumarate (DMF) is a fumaric acid ester used in the treatment of psoriasis and relapsing remitting multiple sclerosis and, while its mechanism of action is unclear, it possesses anti-inflammatory and anti-oxidant properties and also impacts on cell metabolism. Here we treated 16–18-month-old female and male mice with DMF for 1 month and assessed its effect on microglia. The evidence indicates that it exerted sex-specific effects on microglial morphology and metabolism, reducing glycolysis only in microglia from female mice. The data suggest that this may result from its ability to inactivate glyceraldehyde-3-phosphate dehydrogenase (GAPDH).

## 1. Introduction

Uncontrolled microglial activation is a significant factor in the pathogenesis of many neurodegenerative diseases, but it is also known that these cells become activated with age, and this is associated with neuroinflammation and compromised neuronal function. The neuroinflammation is typified by increased expression of markers of microglial activation and production of inflammatory cytokines including IL-1β, IL-6 and TNFα [1].

Sexual dimorphism is recognized in microglial cells with evidence of sex-related differences in the number, phenotype, function and transcriptome of microglia and, in some cases, the differences extend to different brain regions [2,3,4]. The sexual dimorphism is at least partly attributed to the well-accepted protective and anti-inflammatory actions of oestrogens [5]. More recent evidence has suggested that sex-related differences in microglia persist into adulthood [4] with evidence of male mice exhibiting evidence of greater inflammation [2]. With age, this changes and it has been shown that genes which reflect primed or activated microglia are upregulated to a greater extent in females [6]

It has been known for several years that when macrophages adopt an inflammatory phenotype, similar to the age-related changes in microglia, their metabolism shifts towards glycolysis [7]. Recent evidence has indicated this metabolic shift also occurs in inflammatory microglia and data from this lab have demonstrated that activation of microglia from neonatal mice with stimuli like LPS, Aβ and IFNγ results in an increase in glycolysis together with an increase in PFKFB3 a key glycolytic enzyme [8,9]. Similarly, microglia prepared from aged mice [1] and APP/PS1 mice [10], which adopt an inflammatory phenotype, display increased glycolysis and increased PFKFB3 and these changes are associated with reduced phagocytosis and chemotaxis. Interestingly, we have recently found that activation of microglia with the associated switch in metabolism and function that occur in microglia from APP/PS1 mice are sex-dependent, with more profound changes in cells from female, compared with male mice [11].

Dimethyl fumarate (DMF) is a fumaric acid ester used in the treatment of psoriasis and relapsing remitting multiple sclerosis [12,13]. It is rapidly metabolised to MMF to which its biological effects are attributed [14]. The evidence suggests that the beneficial effect of DMF in treating multiple sclerosis and psoriasis derives from its ability to deplete intracellular glutathione (GSH) stores, which ultimately generates type II dendritic cells that produce IL-10 instead of IL-12 and IL-23 [15]. Multiple signalling pathways are activated by DMF and key actions include activation of Nrf2 and inhibition of NFκB [16]. DMF attenuated disease, increasing Nrf2 immunoreactivity in neurons, astrocytes and oligodendrocytes of the motor cortex in the experimental autoimmune encephalomyelitis (EAE) mouse model of MS. It exerted no effect in mice deficient in Nrf2 leading to the conclusion that the cytoprotective effects of DMF are dependent on Nrf2 activation [16]; indeed, these authors showed that Nrf2 activation underpinned the actions of DMF on astrocytes and neurons reducing astrocytic activation and preserving neuronal integrity. However, DMF exerts multiple other effects including inhibition of NFκB, perhaps secondary to Nrf2 activation [17] while it also inhibits glycolysis as a result of succination and inactivation of the glycolytic enzyme, glyceraldehyde-3-phosphate dehydrogenase (GAPDH) [18]. Less is known about the effect of DMF in microglia, although it attenuates LPS-induced inflammatory cytokine production in rat [19] and mouse [20] microglial cell lines. 

The aim of this study was to determine whether DMF modulated microglial phenotype in aged mice and to establish whether any effects were sex-dependent. The data indicated that the treatment of female mice with DMF decreased markers of microglial activation and modulated microglial morphology, but this was not observed in DMF-treated male mice. Furthermore, DMF exerted sex-dependent effects on microglial glycolysis and Nrf2 activation. 

## 2. Materials and Methods

### 2.1. Animals and Treatments

Female and male (aged 16–18 months) C57BL/6 mice were used in these experiments. All experiments were performed under license from the Health Products Regulatory Authority of Ireland in accordance with EU regulations and with local ethical approval (Trinity College Dublin). Animals were housed under controlled conditions (20–22 °C, food and water *ad libitum*) and maintained under veterinary supervision.

Mice were treated daily with DMF (75 mg/kg) or the vehicle (0.08% methylcellulose) for 4 weeks via oral gavage. This is the preferred route of administration and 50–100 mg/kg has been shown to reduce the severity of symptoms in EAE [21,22] and to increase MMF in plasma and brain 30 min after administration [21], while 300 mg/kg reduced injury-induced nociceptive hypersensitivity [23]. Fresh vehicle was prepared every week and fresh DMF was prepared every 2 days. One week before the start of the experiment, animals were handled by the experimenter and given drinking water by oral gavage to familiarise them with the procedure.

At the end of the treatment period, mice were anaesthetised with sodium pentobarbital (Euthanimal) and transcardially perfused with saline. The brain was dissected free and used for preparation of microglia, immunohistochemical analysis, RT-qPCRs, ELISA or enzyme analysis (see below). We undertook analysis in hippocampal and/or cortical tissue because we have previously shown evidence of microglial activation in both areas in aged and APP/PS1 mice [1,10,24] and, in this study, samples were prepared from whole hippocampus and cortex.

### 2.2. Preparation of Microglia 

Tissue was homogenised using the gentleMACS Dissociator (Miltenyi Biotec, Woking, UK) in combination with the AdultBrain Dissociation Kit (Miltenyi Biotec, Woking UK) according to the manufacturer’s instructions. The dissociated tissue was filtered and washed with Dulbecco’s phosphate-buffered saline (D-PBS; PBS containing calcium (100 mg/L), magnesium (100 mg/L), glucose (1000 mg/L), and pyruvate (36 mg/L)). Samples were centrifuged (3000× *g*, 10 min) to remove cell debris and the resultant supernatant was centrifuged (300× *g*, 10 min) to remove red blood cells. The pellet containing the microglia was resuspended in D-PBS, and the microglia were incubated with CD11b microbeads (Miltenyi Biotec. Woking, UK) and magnetically separated using the QuadroMACS separator (Miltenyi Biotec, Woking, UK) according to the manufacturer’s instructions. Samples were resuspended in PBS containing 0.5% foetal bovine serum and centrifuged (300× *g*, 10 min). The final microglial pellet was resuspended in PBS (75 μL) and cells were seeded in 48-well plates (8 × 10^4^ cells/well; final volume 200 μL) or 96-well plates (6 × 10^4^ cells/well; final volume 180 μL) and cultured in Dulbecco’s modified Eagle’s medium (cDMEM) supplemented with macrophage colony stimulating factor (M-CSF; 100 ng/mL; R&D Systems, Minneapolis, MN, USA) and granulocyte macrophage colony stimulating factor (GM-CSF; 100 ng/mL; R&D Systems). The microglia were cultured for 5 days and the media was changed every second day. FACS analysis revealed that this method of preparation yielded samples that were > 95% microglia (Guillot-Sestier et al. unpublished). On day 5, the metabolic profile of cells was assessed using the SeaHorse Extracellular Flux (XF96) Analyser (SeaHorse Bioscience, Billerica, MA, USA), or cells were assessed for immunocytochemistry to analyse PFKFB3 expression. 

### 2.3. Immunocytochemistry 

For immunocytochemistry, cells were washed, incubated with blocking solution (10% BSA, 0.01% Triton in PBS, 30 min), and incubated overnight with rabbit anti-PFKFB3 antibody (1:250; Abcam, Cambridge, UK). Samples were washed and incubated with the secondary antibody (anti-rabbit 546+, 1:1000, 2 h) after which time cells were washed and incubated in the presence of anti-rabbit Alexa Fluor IgG 546+ (2 h; 1:1000; Thermo Fisher Scientific, Warrington, UK). Cells were washed, stained with the nuclear marker DAPI, coverslipped and stored in a dark box at 4 °C until analysis. Images of cells were taken using a Leica SP8 scanning confocal microscope (40X, 5 fields of view), using the same exposure for all the samples and 3D surface images were created for DAPI and PFKFB3 using the same thresholds in 2 separate channels. Images were processed using Imaris (X64 7.6.0) imaging software and PFKFB3 in DAPI-stained nuclei and cytosol was assessed. 

### 2.4. Metabolic Analysis 

Microglia were seeded (6 × 10^4^ cells/well; 180 μL) on SeaHorse 96-well cell culture plates, SeaHorse XF Calibrant solution (200 µL; SeaHorse BioScience, Billerica, MA, USA) was added and cells were incubated overnight (37 °C; CO_2_-free incubator). Cells were washed, assay medium was added (final volume, 180 μL/well) and incubation proceeded in a CO_2_-free incubator (37 °C, 1 h). For the glycolytic flux assay, glucose (10 mM), oligomycin (20 μM) and 2-deoxy-D-glucose (2-DG; 500 mM; Sigma-Aldrich, Gillingham, UK) were prepared in glycolytic flux assay media and loaded into the appropriate ports. Following calibration, the extracellular acidification rate (ECAR) was measured at 8 min intervals over 96 min during which time glucose, oligomycin and 2-DG were injected sequentially at 24 min intervals. ECAR was automatically calculated using the SeaHorse XF96 software and 4–6 replicates were assessed per sample.

### 2.5. Analysis of CD11b mRNA by PCR

CD11b mRNA expression, as well as expression of HO-1, NQO1 and SOD, were assessed in hippocampal samples prepared from DMF- or vehicle-treated mice by RT-PCR. RNA was isolated using the Nucleospin^®^ RNAII KIT (Macherey-Nagel, Duren, Germany) and cDNA was prepared using the High-Capacity cDNA RT kit according to the manufacturer’s instructions (Applied Biosystems, Warrington, UK). Real-time PCR was performed with predesigned Taqman gene expression assays (CD11b: Mm00434455_ml; HO-1; Mm00516005, NQO1; Mm01253561, and SOD; Mm01344233; Applied Biosystems, Warrington, UK) using an Applied Biosystems 7500 Fast Real-Time PCR machine (Applied Biosystems, Warrington, UK). Samples were assayed as previously described (Costello et al., 2016) with β-actin (Mm00407939_s1) as the endogenous control to normalize gene expression data. Gene expression was calculated relative to the endogenous control samples and to the control sample.

### 2.6. Analysis of TNFα and IL-1β by ELISA 

TNFα and IL-1β concentrations were assessed in samples of homogenized cortex prepared from mice in the 4 treatment groups. Briefly, 96-well plates (Nunc-Immuno plate with Maxisorp surface, Søborg, Denmark) were coated with goat anti-mouse TNFα or IL-1β capture antibody (R&D Systems, Minneapolis, MN, USA), incubated overnight at room temperature and blocked with 1% BSA in PBS (200 μL/well; 1 h; room temperature). Duplicate samples and standards were added (50 μL/well) and incubated (2 h; room temperature), plates were washed and samples were incubated with detection antibodies (biotinylated goat anti-mouse TNFα or IL-1β; 1% BSA in PBS; 50 μL/well; 2 h; room temperature), washed and incubated with HRP conjugated streptavidin (50 μL/well; 1% BSA in PBS; 20 min) and washed again. Substrate solution (50 μL/well; 1:1 H_2_O_2_:tetramethylbenzidine; Sigma-Aldrich, Gillingham, UK) was added, incubation proceeded (20 min; room temperature; in darkness) and the reaction was stopped using 1M H_2_SO_4_ (25 μL/well). Absorbance was read at 450 nm with a 540 nm wavelength correction in a BioTek Synergy HT microplate reader. 

### 2.7. Analysis of GAPDH 

GAPDH activity was quantified using a GAPDH activity assay kit (Abcam, Cambridge, UK). Cortical tissue was homogenised in a GAPDH assay buffer (final concentration: 30 μg tissue/100μL buffer) and enzyme activity was assessed in a 96-well plate according to manufacturer’s instructions. Absorbance was measured at 450 nm in kinetic mode at 37 °C. A total of 21 readings were recorded over 40 min and GAPDH activity was calculated according to the manufacturer’s instructions. Protein quantification of lysates was performed using the standard BCA Kit (Pierce, Thermo Fisher Scientific, Warrington, UK), and data are expressed as U/mg of total protein. 

### 2.8. Immunohistochemistry

Brain tissue was fixed in 4% paraformaldehyde for 24 h for later preparation of sections for immunohistochemical analysis. Following fixation, the tissue was incubated in 30% sucrose (48 h), embedded in an optimal cutting temperature (OCT) solution and frozen, using a dry ice plaque, in a cryomold. Sections (25 μm or 40 μm) were stored at −20 °C in a freezing solution (30% ethylene glycol, 30% sucrose in PBS) until required for immunohistochemistry.

For p16 staining, the tissue was incubated with citrate buffer (20 min, 85 °C) to facilitate antigen retrieval. Samples were washed, and the tissue was permeabilised (10 min; PBT: PBS with 0.1% Triton-X100), blocked (1 h; PBT with 3% BSA) and incubated (overnight, 4 °C in primary antibodies (rabbit anti-Iba-1; 1:750; Wako, Japan and mouse anti-CDKN2A/p16^INK4a^; 1:200; Abcam, Cambridge, UK). Samples were washed and incubated with secondary antibodies (Alexa Fluor 594+ anti-rabbit and Alexa Fluor 488+ anti-mouse; Thermo Fisher Scientific, Warrington, UK) for 2 h in the dark at RT on a plate shaker. 

For assessment of Nrf2 and p65 NFκB translocation to the nucleus of Iba1^+^ microglia, sections were incubated in PBS (1 h) in wells of a 24-well plate. Samples were washed, and a blocking buffer (phosphate-buffered saline plus 0.3% Triton X-100 containing 10% BSA) was added before incubation continued (1 h on a plate shaker). Sections were washed and the primary antibody (250 μL; 4 °C; overnight; Table 1) was added and the plate was placed in a cold room on a plate shaker overnight. The sections were washed, the appropriate secondary antibody (1:1000; 250 μL, 2 h; Table 1) was added, and incubation continued (2 h; RT in the dark). 

Following a final wash, sections were mounted on slides using an Antifade mounting medium with DAPI (Vector Laboratories, Inc). Slides were left to dry in the dark and were stored in a dark box at 4 °C until further analysis. Images were taken using a fluorescence microscope (Leica SP8 microscope, LACx programme, 40X, 9 fields of view per section) using the same exposure and conditions for all samples. ImageJ software (National Institute of Health, http://imagej.nih.gov/ij/; accessed 20 November 2021) was used to process the images. Iba1 is expressed as the integrated density (IntDen). Using ImageJ software, positive staining (i.e., fluorescence) was converted into pixels and IntDen (area x mean grey value), which is an indirect assessment of Iba1 expression, was calculated. 

To evaluate activation of Nrf2 and NFκB, the number of DAPI-stained Iba1^+^Nrf2^+^ cells and DAPI-stained Iba1^+^p65 NFκB^+^ cells, and the corresponding total number of Iba1^+^ cells for the 2 separate markers, were manually counted using the ImageJ cell counter tool. Data are presented as nuclear Iba1^+^Nrf2^+^ cells and Iba1^+^p65 NFκB^+^ cells as a percentage of the corresponding total Iba1^+^ cells. Similarly, the number of Iba1^+^p16^+^ cells was calculated as a percentage of the total Iba1^+^ cells.

### 2.9. Analysis of Microglial Morphology 

Image stacks were taken from Iba1-stained hippocampal tissue sections (40 μm) using a Leica SP8 scanning confocal microscope, and image stacks were converted to 3D images with the surface-rendering feature of Imaris BitPlane software (version 7.6.5). Images were converted to 8-bit greyscale using ImageJ (National Institute of Health, https://imagej.nih.gov/ij/; accessed 20 November 2021), a threshold was set and binarized images were filtered by pixel size to reduce background and enhance contrast. Soma diameter, perimeter and size were assessed, and the process number and length, as well as the number of junctions and triple and quadruple points/cell, were quantified using the Analyze Skeleton plugin (Morisson et al., 2017).

### 2.10. Statistical Analysis

Data are reported as the mean ± SEM and the number of experiments is indicated in each case. Statistical analysis was carried out using a 2-way ANOVA followed by the Tukey post hoc test. The significance level was set at *p* < 0.05.

## 3. Results

### 3.1. DMF Differentially Affects Markers of Microglial Activation in A Sex-Dependent Manner

We assessed the effect of DMF treatment in sections from 16–18-month-old male and female mice and report that DMF differentially affected microglia in a sex-dependent manner. Significant sex x treatment interactions were observed in two markers of microglial activation. Iba1 staining intensity (*p* < 0.01; Figure 1A,B) and CD11b mRNA (*p* < 0.05; 2-way ANOVA; Figure 1C) and post hoc analysis indicated that both measures were decreased in the tissue from female DMF-treated mice compared with female vehicle-treated mice (* *p* < 0.05; ** *p* < 0.01) and male DMF-treated mice (^+^
*p* < 0.05; ^++^
*p* < 0.01).

One measure of cell senescence is p16, a cell cycle inhibitor that accumulates in senescent cells [25] and is increased in the brain of aged mice [1] and the data showed that the effect of DMF on the proportion of p16^+^ Iba1^+^ cells was sex-specific (*p* < 0.05; significant main effect of sex; Figure 1D,E). The proportion of p16^+^ Iba1^+^ cells was decreased in DMF-treated female, but not male, mice and post hoc analysis indicated that the proportion of p16^+^ Iba1^+^ cells was significantly reduced in DMF-treated female, compared with male, mice (^+^
*p* < 0.05).

We then asked whether DMF might also affect microglial morphology differently in male and female mice and the data showed that there was a significant sex x treatment interaction in soma diameter (*p* < 0.01), and perimeter, size, density and circularity (*p* < 0.001; Figure 2A–D). Post hoc analysis revealed that soma diameter, perimeter and size were increased in the tissue from female DMF-treated mice compared with female vehicle-treated mice (** *p* < 0.01; *** *p* < 0.001) and male DMF-treated mice (^+^
*p* < 0.05; ^++^
*p* < 0.01).

DMF also increased the complexity of microglia, albeit only in female mice (Figure 2E–I); thus there was a significant sex x treatment interaction in the number (*p* < 0.001) and length (*p* < 0.01) of processes, and the number of junctions (*p* < 0.001), and number of triple (*p* < 0.01) and quadruple (*p* < 0.001) points. There was a significant increase in the number of processes, junctions, triple and quadruple points in microglia from DMF-treated female mice compared with vehicle-treated female mice (*** *p* < 0.001) and male DMF-treated mice (^+++^
*p* < 0.001). In contrast, there was a significant decrease in the length of processes in microglia from DMF-treated female mice compared with vehicle-treated female mice (*** *p* < 0.001) and male DMF-treated mice (^+++^
*p* < 0.001).

### 3.2. DMF Impacts on Microglial Metabolism in A Sex-Dependent Manner

Previous data have determined that microglia express increased Iba1 and/or that CD11b adopt a glycolytic phenotype [1] and here, analysis of the metabolic profile of cells, indicated that ECAR was decreased in microglia prepared from DMF-treated, compared with vehicle-treated, female mice (Figure 3A), whereas there was an increase in ECAR in microglia prepared from DMF-treated male mice compared with vehicle-treated male mice. There was a significant sex x treatment interaction in both glycolysis and glycolytic capacity (*p* < 0.001; Figure 3B,C) and post hoc analysis revealed that DMF significantly decreased both measures in microglia from female mice (* *p* < 0.05; Figure 3B,C) while it increased both in microglia from male mice (* *p* < 0.05). A significant difference in glycolysis and glycolytic capacity was also observed between microglia from DMF-treated female mice compared with DMF-treated male mice (^++^
*p* < 0.01).

PFKFB3 is a master regulator of glycolysis [26,27], the expression of which correlates with microglial activation in aged mice and APP/PS1 mice [1,8,10,11]. We assessed cytosolic PFKFB3 as a proxy marker of enzyme activation and, although no statistically significant changes were observed between treatment groups, DMF decreased cytosolic PFKFB3 in microglia from female mice by 40% which approached significance (*p* = 0.081), while no difference in total PFKFB3 was observed (Figure 3F).

It has been proposed that the DMF-induced change in glycolysis derives from its ability to inactivate a key glycolytic enzyme, GAPDH [18], that converts D-glyceraldehyde 3-phosphate into 3-phospho-D-glyceroyl phosphate. We assessed its activity in samples of cortical homogenate prepared from mice in each of the treatment groups. The data indicated that there was a significant sex x treatment interaction (*p* < 0.001; Figure 3G) and post hoc analysis revealed that GAPDH activity was significantly decreased in tissue from female DMF-treated mice compared with female vehicle-treated mice (* *p* < 0.05). The data also showed that enzyme activity was decreased in vehicle-treated male mice compared with vehicle-treated female mice (^+++^
*p* < 0.001), which reflected the difference in ECAR between male and female vehicle-treated mice shown in Figure 3A.

### 3.3. DMF Increases Activation of NFkB and Nrf2

We assessed whether the change in microglial activation, morphology and metabolism might be linked with reported changes in the activation of NFκB or Nrf2, which are transcription factors that underpin inflammatory processes and anti-oxidative processes, respectively. As a measure of NFκB activation, we analysed p65 in the nucleus of Iba1^+^ cells in sections of hippocampus from DMF- and vehicle-treated mice. A significant main effect of DMF was observed (*p* < 0.001; Figure 4A) and post hoc analysis indicated that the expression of nuclear NFκB in microglia was significantly decreased in sections from DMF-treated female mice compared with control females and also DMF-treated male mice compared with control males (*** *p* < 0.001; Figure 4B). No difference in nuclear NFκB expression was observed in DMF-treated female compared with DMF-treated male mice. This finding was replicated in cortical tissue (Appendix A). DMF also decreased hippocampal concentrations of TNFα and IL-1β (*p* < 0.001; Figure 4C,D) and post hoc analysis indicated significant decreases in tissue from DMF-treated female and male mice compared with their respective vehicle-treated counterparts (* *p* < 0.05; *** *p* < 0.001) suggesting that the modulatory effect of DMF on secretory function of microglia is not affected by sex. We assessed the engulfment of myelin oligodendrocyte glycoprotein as an indicator of phagocytosis and found that there was no significant difference between groups (data not shown).

The data also indicated that DMF drives Nrf2 translocation to the nucleus. There was a significant main effect of DMF on Nrf2 (*p* < 0.001) and post hoc analysis indicated that there was a significant increase in Nrf2 staining in the nucleus of Iba1^+^ cells prepared from DMF-treated female and male mice compared with their respective vehicle-treated controls (** *p* < 0.01; *** *p* < 0.001; Figure 5A,B). Interestingly, in cortical tissue, the effect of DMF on Nrf2 translocation to the nucleus was confined to female mice (*** *p* < 0.001; Appendix A) where a significant difference between nuclear Nrf2 was observed between DMF-treated male and female mice (^+^
*p* < 0.05). The DMF-induced change in Nrf2 activation in the hippocampus did not translate into parallel changes in mRNA expression of antioxidants. The data indicated that DMF exerted no significant effect on mRNA expression of HO-1 or NQO1 (Figure 5C,D) although a significant main effect of DMF was observed in SOD mRNA (*p* < 0.05; Figure 5E) and DMF decreased SOD mRNA in hippocampal tissue from males (* *p* < 0.05). A significant main effect of sex was observed in HO-1 mRNA (*p* < 0.001; Figure 5C) and mRNA expression of HO-1 was significantly reduced in the hippocampal tissue from vehicle-treated males compared with vehicle-treated females (^+^
*p* < 0.05) and from DMF-treated males compared with DMF-treated females (^+^
*p* < 0.05). A significant main effect of sex was observed in HO-1 mRNA in cortex (*p* < 0.05; Appendix A) and a significant main effect of DMF was observed in SOD mRNA in cortical tissue (*p* < 0.05; Appendix A).

## 4. Discussion

The data indicate that DMF exerts a sex-dependent effect on microglia altering their activation state, metabolism and morphology; these actions of DMF were confined to female mice. In addition, DMF decreased glycolysis in microglia from female mice perhaps because it activates GAPDH and decreases PFKFB3.

It is well established that markers of microglial activation, for example CD68, CD11b and Iba1, increase with age but there is no clear evidence in the literature of a sex-related difference in expression of these markers. Here, assessment of CD11b mRNA and Iba1 immunoreactivity, also did not reveal differences between 16–18-month-old male and female vehicle-treated control mice. However, microarray analysis in hippocampal tissue prepared from young, middle-aged and old mice showed that there was a greater age-related shift in expression of genes that reflect inflammatory pathways in female, compared with male, mice [28]. Our evidence indicates that DMF exerted sex-related effects, decreasing CD11b mRNA and Iba1 immunoreactivity in females only; we are unaware of any previous reports of sexual dimorphism with respect to DMF effects. However, DMF has been shown to modulate microglial activation in vitro and in vivo. For instance, it attenuates LPS-induced changes in microglial cell lines [19,20] and primary microglia [29], but our unpublished data indicate that neither sex-related or DMF-associated change were observed on ECAR in cultured microglia (data not shown). In addition to the effects on LPS-induced changes in vitro, DMF also suppresses microglial activation in the hippocampus induced by a peripheral LPS challenge in mice [29] and decreased microglial activation in the spinal cord [19] and hippocampus [30] of streptozotocin-treated animals. These latter changes were described in young mice and no sex-related differences were reported. However, it remains to be determined whether the changes extend to other circumstances in which microglial activation has been identified and/or when sex-related differences occur, for example, when the microbiome is disturbed [31].

In the past few years, research has begun to focus on evaluating sex-related differences in microglia with clear differences identified during development and early life [32]. There is growing evidence of changes that persist into adulthood and age, including increased antigen presentation in the cortex of males compared with females [4], upregulation in microglia-specific genes that are indicative of inflammatory processes in aged females compared with males [33], particularly aged female APP/PS1 mice [11], increased responsiveness to LPS in aged female mice compared with males [34], sex-related responses to engrafted microglia in EAE [35] and middle cerebral artery occlusion [2], and sex-related differences in response to trauma [36]. The factors that contribute to these changes remain to be clarified but probably include hormonal differences [5] and epigenetic changes [37].

Our data show that DMF also exerted a marked sex-related effect on microglial morphology, decreasing soma size, perimeter and diameter and increasing the number and complexity of processes, but this was observed only in female mice with no discernible effects in males. Thus, in addition to the sex-specific effects on CD11b mRNA and Iba1 immunoreactivity, DMF also exhibited sexual dimorphism in microglial complexity. A decrease in microglial complexity, and specifically a reduction in the number of microglial processes, has been reported following brain injury [38] and with age [39] although, to our knowledge, no sex-related differences have been reported.

We have linked changes in microglial activation and phenotype with changes in the metabolic profile of cells in previous studies. Specifically, as in the case of macrophages [7,40], activated microglia in the aged brain and brain of APP/PS1 mice switch their metabolism towards glycolysis [1,8,9,10]. Furthermore, in microglia from APP/PS1 mice, microglial activation was more marked, and the switch to glycolysis was observed only in cells from female mice [11]. Here, the DMF-induced changes in microglia were accompanied by metabolic changes; the data show that DMF decreases glycolysis in microglia isolated from female mice while the opposite effect of DMF was observed in cells from male mice where glycolysis was increased. The link between inflammation and glycolysis is well established in macrophages [7] and cultured microglia stimulated with IFNγ- and LPS + Aβ are glycolytic and produce inflammatory cytokines [8,9], while microglia isolated from aged mice [1] and APP/PS1 [11] mice exhibit markers of activation and switch metabolism towards glycolysis. Here, parallel DMF-induced decreases in markers of microglial activation and glycolysis were observed in samples from female mice. We did not observe a decrease in TNFα or IL-1β and, although this was not directly assessed, it is possible that this is because cytokine concentration was assessed in hippocampal homogenate rather than in supernatant samples obtained from isolated microglia.

DMF exerts a plethora of effects, one of which is to cause the inactivation of GAPDH, which catalyses the conversion of glyceraldehyde 3-phosphate to 1,3-biphosphoglycerate during glycolysis. It does so by covalently modifying the enzyme as a consequence of succination of a cysteine residue at the active site resulting in the inhibition of glycolysis in macrophages and lymphocytes [18]. The data presented indicate that DMF-induced changes in GAPDH activity, which mirrored the changes in glycolysis, decreases its activity exclusively in the tissue from female mice and suggests that this action underpins the effect on glycolysis. In contrast, DMF increased GAPDH activity in the tissue from male mice, perhaps explaining the DMF-induced increase in glycolysis.

We have shown previously that glycolytic cells exhibit an increase in PFKFB3 and have proposed that its upregulation may trigger microglia to adopt a glycolytic phenotype [1,8,9,10]. Consistent with this is the observation that inhibition of PFKFB3 by 3PO blocked the LPS + Aβ-stimulated switch to glycolysis in microglia [40]. Here we assessed the translocation of PFKFB3 from the nucleus to the cytosol as a measure of enzyme activity and showed that translocation was decreased in cells prepared from DMF-treated females by about 40%, albeit insignificantly, but there was no change in cells from male mice.

Among the many cellular events triggered by DMF are decreased NFκB activation and the present findings shows that DMF decreases NFκB activation in Iba1^+^ cells, although the effect was similar in male and female mice and therefore is unlikely to be key to the sexual dimorphism observed in relation to the effects of DMF on microglial morphology. DMF also decreased TNFα and IL-1β in hippocampus, albeit to a similar extent in samples from males and females. This is broadly consistent with previous findings indicating that DMF decreases inflammatory cytokine production in LPS-treated microglia [41] and microglial cell lines [19,20].

DMF also triggers activation of Nrf2, which is considered to be the master regulator of cellular antioxidant responses since its activation controls the transcription of genes that target redox homeostasis [42]. Nrf2 also modulates several other cell activities [43] including inhibition of NFκB activation by phosphorylating IκB [13], and therefore it can modulate inflammation [29]. Ageing has been associated with compromised Nrf2 activation [44] and here, we showed that the administration of DMF increased Nrf2 activation in microglia, although this did not translate into parallel changes in the antioxidants assessed here. Interestingly, whereas there was no evidence of a sex-related change in Nrf2 activation in the hippocampus, a sex-related difference was observed in cortical tissue where DMF increased Nrf2 activation in the microglia of female mice only.

A role for Nrf2 activation in modulating microglial activation has been described. Thus, the transfection of neonatal microglia with Nrf2-specific siRNA increases the production of TNFα and IL-6, whereas the overexpression of Nrf2 had the opposite effect [45]. Similarly, Nrf2-deficient mice are hypersensitive to the inflammatory effects induced by MPTP [46] and LPS [47], resulting in an upregulation of markers of microglial activation. However, even in Nrf2-deficient microglia, DMF attenuated LPS-induced upregulation of inflammatory cytokines and p65 NFκB translocation to the nucleus, albeit to a lesser extent, indicating that these changes appeared not to be driven entirely by the activation of Nrf2 [29]. The results from the present experiment suggest that the DMF-induced change in Nrf2 activation in microglia is unlikely to impact on their metabolic profile.

## 5. Conclusions

The data presented add to the limited findings describing the effect of DMF on microglia but, most importantly, demonstrate that DMF acts on microglia in a sex-specific manner and attenuates the age-related metabolic change in cells from female mice, probably because it inactivates GAPDH. The question arises whether DMF exerts sex-related differences in humans. We are unaware of any data describing sex-related differences in the effect of DMF in psoriasis, while two studies which have assessed the efficacy of treatments including DMF in MS, did not report on sex-related differences [48,49], although it has been shown that gastrointestinal tolerability for DMF is reduced in females [50].

The importance of the present findings lies in the fact that DMF can modify microglial activation, which is a characteristic of MS [51] and it is significant that it attenuates microglial activation in a mouse model of the disease [20].

## Figures and Tables

**Figure 1 cells-11-00729-f001:**
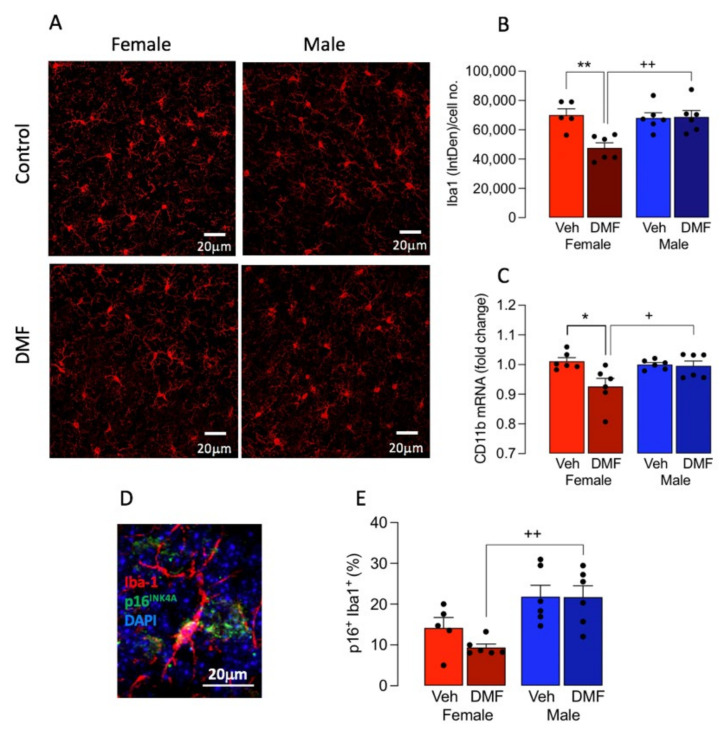
DMF decreased microglial activation in female mice. (**A**) Iba1 staining was performed using immunohistochemistry in tissue sections from DMF- and vehicle-treated male and female mice. The sample immunograph suggests that DMF decreases Iba1 immunoreactivity in sections prepared from female mice. Scale bar = 20 μm. (**B**) Analysis of the mean data revealed a significant treatment x sex interaction (*p* < 0.01; 2-way ANOVA) and post-hoc analysis indicated that Iba1 immunoreactivity was significantly decreased in sections from DMF-treated, compared with vehicle-treated control, female mice (* *p* < 0.05) and compared with DMF-treated male mice (^++^
*p* < 0.01). Data are represented as means ± SEM (*n* = 5 or 6). (**C**) Analysis of mRNA expression of CD11b in tissue prepared from the 4 groups of mice revealed a significant treatment x sex interaction (*p* < 0.05; 2-way ANOVA) and post-hoc analysis indicated that CD11b mRNA was significantly decreased in tissue from DMF-treated, compared with vehicle-treated, female mice (* *p* < 0.05) and compared with DMF-treated male mice (^+^
*p* < 0.05). Data are represented as means ± SEM (*n* = 5 or 6). (**D**,**E**). Analysis of p16^+^ Iba1^+^ cells in sections prepared from male and female DMF-treated and vehicle-treated mice revealed that there was a significant main effect of sex (*p* < 0.001; 2-way ANOVA) and post hoc analysis indicated that there was a significant reduction in p16^+^ Iba1^+^ cells in tissue from DMF-treated female, compared with DMF-treated male, mice (^++^
*p* < 0.01). Data are represented as p16^+^ Iba1^+^ cells expressed as a percentage of Iba1+ cells and expressed as the mean ± SEM (*n* = 5 or 6; 3 sections/mouse, 3 fields of view/section; approximately 400 cells/mouse). The inset shows a typical p16+ Iba1+ cell. Scale bar = 20 μm.

**Figure 2 cells-11-00729-f002:**
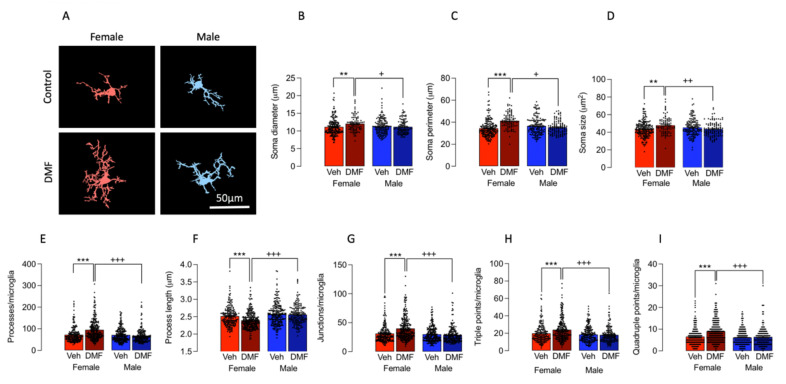
DMF impacts on microglial morphology in a sex-dependent manner. (**A**) Masks were prepared for analysis of morphological features of microglia. Scale bar = 50 μm. (**B**–**D**) Analysis of the data for soma diameter, perimeter and size revealed significant treatment x sex interactions (*p* < 0.01 for diameter and *p* < 0.001 for perimeter and size) and post hoc analysis indicated that there was a significant increase in each measure in tissue from DMF-treated, compared with vehicle-treated, female mice (** *p* < 0.01; *** *p* < 0.001) and compared with DMF-treated male mice (^+^
*p* < 0.05; ^++^
*p* < 0.01). (**E**–**I**) Analysis of process number and length, and numbers of junctions, triple and quadruple points indicated that there was a significant treatment x sex interaction in all cases (*p* < 0.01 for process length and *p* < 0.001 for the others). Post hoc analysis revealed that DMF significantly increased process number and numbers of junctions, triple and quadruple points in sections from female mice compared with vehicle-treated control female mice (*** *p* < 0.001) and DMF-treated male mice (^+++^
*p* < 0.001). In contrast, the process length was significantly decreased in DMF-treated females compared with vehicle-treated control female mice (*** *p* < 0.001) and DMF-treated male mice (^+++^
*p* < 0.001). Data from 1 experiment are represented as means ± SEM derived from 4–7 mice with 18–51 cells counted per mouse.

**Figure 3 cells-11-00729-f003:**
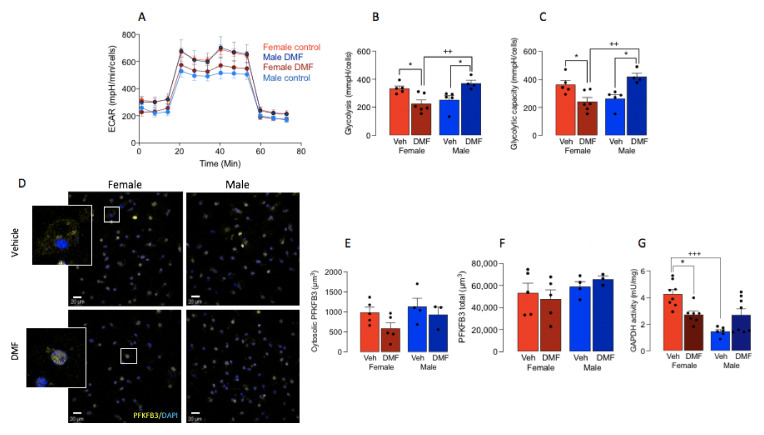
DMF modulates microglial metabolism. (**A**) Analysis of the metabolic profile of microglia isolated from DMF- and control-treated mice indicated that DMF decreased ECAR in cells from females and increased ECAR in cells from males. (**B**,**C**) Analysis of glycolysis and glycolytic capacity indicated significant treatment x sex interactions (*p* < 0.001) and post hoc analysis indicated that both values were significantly decreased in microglia prepared from DMF-treated female, compared with vehicle-treated female mice (* *p* < 0.03) and DMF-treated male, mice (^++^
*p* < 0.01). Glycolysis and glycolytic capacity were significantly increased in cells from DMF-treated male mice, compared with vehicle-treated male mice (* *p* < 0.05). (**D**,**E**) PFKFB3 (yellow) assessed in isolated microglia from DMF- and vehicle-treated male and female mice. The sample immunograph suggests that DMF increased cytosolic staining of PFKFB3 in female, but not male, mice and increased translocation to the DAPI-stained nucleus (blue). The mean data (**E**), 3 coverslips/mouse, average number of cells counted/coverslip = 48) indicate a clear difference between treatment groups in females but no statistically significant changes were observed. No change in total PFKFB3 (**F**) staining was observed. Scale bar = 20 μm. (**G**) Analysis of GAPDH activity revealed that there was a significant sex x treatment interaction (*p* < 0.001) and post hoc analysis indicated a significant decrease in tissue from female DMF-treated mice compared with female vehicle-treated mice (* *p* < 0.05) and DMF-treated male mice (^+++^
*p* < 0.001). Data from 2 or 3 experiments are presented as means ± SEM (*n* = 3–6).

**Figure 4 cells-11-00729-f004:**
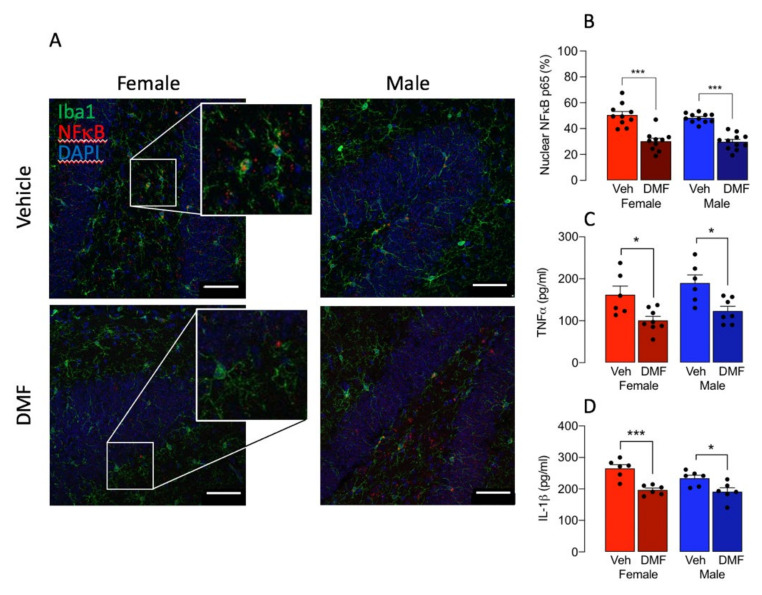
DMF decreases translocation of NFκB to the nucleus in Iba1^+^ microglia in hippocampus. A,B. p65 NFκB expression in DAPI-stained nuclei was assessed in Iba1^+^ microglia in hippocampal sections prepared from DMF- and vehicle-treated control male and female mice. The sample immunographs and insets (**A**) indicate that DMF decreased NFκB staining in sections from male and female mice and the higher magnification insets show increased co-localization of NFκB in DAPI-stained nuclei in control- compared with DMF-treated samples from female mice. Analysis of the mean data (**B**) revealed that there was a significant main effect of treatment (*p* < 0.001; 2-way ANOVA) and post hoc analysis indicated that p65 NFκB^+^ DAPI^+^ staining in Iba1^+^ cells was significantly decreased in sections from DMF-treated male and female mice compared with the respective controls (*** *p* < 0.001). Data are expressed as a percentage of the total number of Iba1^+^ microglia and are presented as means ± SEM (*n* = 10 or 11). Scale bar = 50 μm. (**C**,**D**) A significant main effect of DMF was observed on hippocampal concentrations of TNFα and IL-1β (*** *p* < 0.001) and post hoc analysis indicated significant decreases in tissue from DMF-treated female and male mice compared with their respective vehicle-treated counterparts (* *p* < 0.05; *** *p* < 0.001). Data from 2 experiments are presented as means ± SEM (*n* = 5 or 6).

**Figure 5 cells-11-00729-f005:**
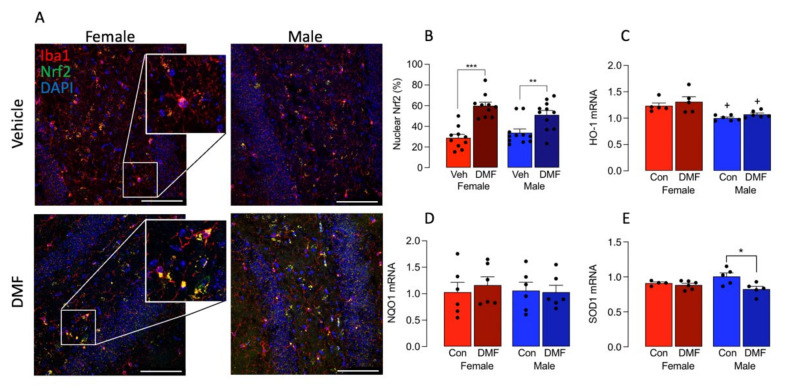
DMF increases the translocation of Nrf2 to the nucleus in Iba1^+^ microglia in hippocampus. (**A**,**B**). Nrf2 expression in DAPI-stained nuclei was assessed in Iba1^+^ microglia in hippocampal sections prepared from DMF- and vehicle-treated control male and female mice. The sample immunographs and insets indicate that DMF increased Nrf2 staining with a significant main effect of treatment (*p* < 0.001; 2-way ANOVA) and the higher magnification insets show decreased co-localization of Nrf2 in DAPI-stained nuclei in control compared with DMF-treated samples from female mice. Post hoc analysis indicated that DMF significantly increased Nrf2^+^ DAPI^+^ staining in Iba1^+^ cells from female and male mice compared with their respective vehicle-treated controls (** *p* < 0.01; *** *p* < 0.001). Data from 2 experiments are expressed as a percentage of the total number of Iba1^+^ microglia and are presented as means ± SEM (*n* = 10 or 11). Scale bar = 50 μm. (**C**) A significant main effect of sex was observed in HO-1 mRNA (*p* < 0.001) and expression was significantly reduced in the hippocampal tissue from vehicle-treated males compared with vehicle-treated females (^+^
*p* < 0.05) and from DMF-treated males compared with DMF-treated females (^+^
*p* < 0.05). (**D**) DMF exerted no significant effect on mRNA expression of NQO1. (**E**) A significant main effect of DMF was observed in SOD mRNA (*p* < 0.05) and post hoc analysis indicated a significant decrease in samples from DMF-treated male mice compared with vehicle-treated male mice (* *p* < 0.05). Data (**C**–**E**) are expressed as means ± SEM (*n* = 5 or 6).

**Table 1 cells-11-00729-t001:** Antibodies used in immunohistochemistry.

Protein	Primary Antibody	Secondary Antibody
Iba1 (for NFκB)	Goat anti-Iba1 (LSBio Inc, Seattle, WA, USA) 1:750	Alexa Fluor+ anti-goat (Thermo Fisher Scientific, Warrington, UK)
NFκB p65	Rabbit anti-NFκB (Santa Cruz Biotech, Dallas, TX, USA); 1:250	Alexa Fluor+ 594+ anti-rabbit (Thermo Fisher Scientific, Warrington, UK)
Iba1 (for Nrf2)	Goat anti-Iba1 (LSBio Inc, Seattle, WA., USA) 1: 500	Alexa Fluor+ 594+ anti-goat (Thermo Fisher Scientific, Warrington, UK)
Nrf2	Rabbit anti-Nrf2 (Santa Cruz Biotech, Dallas, TX, USA); 1:250	Alexa Fluor+ 488+ Donkey anti-rabbit (Thermo Fisher Scientific, Warrington, UK)

## Data Availability

All data will be made available on request.

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
