# Peer review of "The Modulatory Effects of DMF on Microglia in Aged Mice Are Sex-Specific"

_cells, 2022, doi:10.3390/cells11040729_

Round 1

Reviewer 1 Report

Minor concerns:

Line 35: reference is missing.

Major concerns:

Figure 3: n=3 is to low, 3 animals are not representative. You have to increase the n number.

You proposed sex differences in the metabolic profile of microglia, however you could not show any sex differences in the inflammatory response.

M1 macrophages (pro-inflammatory) use glycolysis, while M2 macrophages prefere oxidative phosphorylation.

In figure 4 you showed a sex-independent decline of pro-inflammatory markers after DMF treatment and in figure 3 you showed decreased glycolysis and glycolytic capacity in females and an increase in males. Your results are contradictory. Please discuss your results.

Reviewer 2 Report

            The manuscript submitted by Mela et al descries the sex-specific effect of DMF on microglial phenotype in aged mice. The authors treat aged male and female mice with DMF for one month, and evaluate the microglia in vivo and after culture. They find microglia from aged female mice treated with DMF uniquely alter their activation state, reflected by decreases in Iba1 and CD11b expression and altered morphology. The effects of DMF on microglial metabolism are completely opposite in aged male and female mice. Interestingly, male and female mice treated with DMF exhibit comparable changes in NFkB and Nrf2. This is an interesting study, attempting to provide insight into an area with limited findings thus far. However, a number of issues need to be addressed in order to strengthen these findings and their interpretations. General and specific comments follow below.

General comments:

  1. The idea that DMF, and really any therapeutic, may have sex-specific effects is of critical importance. The sexual dimorphism in microglia and their ensuing inflammatory processes contributing to neurodegenerative diseases is rapidly becoming appreciated and understanding whether and how drugs act in sex-specific ways is an important gap in our knowledge. As it pertains to DMF, the authors provide no discussion or evidence for sex-specific effects of DMF generally, or in any other context (MS, psoriasis, etc). There is some allusion to it in the discussion (lines 400-405), but it stops short of discussing it specifically.

  1. The authors note the recent understanding of the regional specificity of microglia. In this manuscript, most of their work centers on the hippocampus; however, there is no rationale given for focusing on the hippocampus, and there is no clarification of which hippocampal regions were assessed (entire region versus CA1, CA3, dentate gyrus, etc). Some cortical data is included as well, but again, no rationale was given for why that was included, nor which cortical region was assessed. Furthermore, there was no comment or discussion on region-specific differences.

  1. In general, the manuscript lacks sufficient detail in the methods and text to draw meaningful conclusions.
    1. There is no description of the purity of microglial pellet preparation (section 2.2) with implications for interpretation of immunocytochemistry and metabolic analysis.
    2. Many different molecules were assessed (e.g., p16, PFKPF3) with no description of what they are, how they function in the context they’re being assessed, and what the findings mean.
    3. There is no indication of how many individual cells were counted, and how many cells per animal were assessed. One assumes “n=5-6” in Figure 1 refers to individual mice, but there’s no indication of how many cells per mouse. While Figure 2 illustrates a large number of data points, its unclear how many different mice those individual cells presumably came from.

  1. The quality of the figure images, particularly Figures 4 and 5, needs improvement. It is difficult to see the nuclear p65 in Figure 4, and the panels in Figures 4 and 5 do not appear to be of comparable quality. Each contains a panel that is more grainy (over-processed) than the others, leading to questions about image processing.

Specific comments:

  1. The reference(s) for neuroinflammatory markers in lines 34 and 35 need to be added to replace “(REF)”.

  1. In line 59, “GSH” should be spelled out the first time it is mentioned.

  1. There is no description for how immunofluorescent intensity was normalized or variations in background intensity were accounted for in the quantification/analysis.

  1. The scale bars in the figures and figure legends are inconsistent/inaccurate. For example, in Figure 1, the images show “20αm” while the legend says “20mm”, and in Figures 4 and, the images show “20αm” or “20μm” and the legend says “Scale bars = 1mm and 0.5mm in main images and insets respectively”, when the insets do not even include scale bars.

  1. There are no scale bars included for Figure 2.

  1. Related to Figure 3, an assessment of total PFKFB3 levels is necessary to show the changes in its localization are due to trafficking and not changes in overall expression.

  1. There are no labels for the panels in Figure 4A.

  1. Figure 5 lacks a description of each channel (color) in the legend and the images.

  1. There is a change in the symbol used to denote statistical significance in Figure 5, but the reason is unclear. In previous images, “*” was used to denote significant differences with treatment, while “+” denoted significant difference between sexes. Why “”is now used (instead of “+”) is unclear.

  1. There are no representative images for 8OHdG immunostaining in Figure 5.

  1. Granted it is appreciated microglial morphology changes with age, but the statement regarding DMF in age-related microglial complexity (lines 410-411) is a bit of an overstatement, as the authors didn’t show age-related changes in microglial morphology, just morphology at a single point in time in aged mice.

Reviewer 3 Report

In this study, the authors found that Dimethylfumarate (DMF) has a sex-specific effect on microglia in aged mice. DMF exerted sex-specific effects on microglial morphology and metabolism, reducing glycolysis only in microglia from female mice. This topic is interesting. Some concerns and suggestions are listed as below:

  1. What did you mean by saying ‘REF’ in line 35?
  2. In the method, the authors said that mice were treated daily with DMF (75 mg/kg) or the vehicle for 4 weeks via oral gavage. I wonder if there are any other approaches to reduce the times of oral gavage? How about i.p. injections? Why DMF (75 mg/kg) was treated for 4 weeks? Why the dose of DMF (75 mg/kg) was selected? How about a relatively short time period?
  3. Can DMF be found in the brain? The levels of DMF in the circulation and brain (after 4 weeks) should be measured in both male and female mice. This is a major concern.
  4. In Figure 1, Scale bar = 20mm. Please have a check in Figure 1A and D (20 ?m). In addition, different brain regions of staining must be provided.
  5. Apart from CD11b and Iba1, other markers of microglia (M1 or M2) should also be tested.
  6. It is not clear for readers if more microglial proliferation or less microglial apoptosis occurs in male mice within a given brain region than in females?
  7. Since the targets of DMF is broad, what about the effects of DMF on circulating monocytes in female and male aged (16-18 months) C57BL/6 mice?
  8. Apart from microglia, what about the effects of DMF on other types of glial cells such as astrocytes and oligodendrocyte in female and male aged (16-18 months) C57BL/6 mice? Additional experiments are needed.
  9. FACS should be performed to investigate if overall numbers of microglia (or infiltrating macrophages) are sex different in aged mice (and after the treatment of DMF).
  10. The authors found that the expression of nuclear NF-κB expression in microglia was significantly decreased in sections from DMF-treated female and male mice compared with their respective controls (Figure 4). Why the subtitle of this section is ‘DMF increases activation of NF-kB’?
  11. Are there any differences of p65 NF-κB expression between male and female treated with DMF in Figure 4B? Please explain.
  12. It is not clear for readers if these sex differences may still be noted in young or adult mice after DMF treatment. When they became sex-specific differences after treatment? Some reported that microglia exert a sex-specific effect on the long-term absence of the microbiome, with males being significantly affected during early development while females showed profound changes at adult periods instead (Microbiome influences prenatal and adult microglia in a sex-specific manner. Cell. 2018). This point should not be ignored.
  13. Male microglia exhibit higher potential antigen presentation ability in the cortex, as evidenced by higher expression of major histocompatibility complex (MHC) I and MHC II than do age-matched female microglia (Transcriptional and translational differences of microglia from male and female brains. Cell Rep. 2018). This point should be discussed.
  14. Sex-specific effects of microglia-like cell engraftment have been reported and an underestimated yet marked sex-dependent microglial activation pattern may exist in the injured CNS (Sex-Specific Effects of Microglia-Like Cell Engraftment during Experimental Autoimmune Encephalomyelitis, International Journal of Molecular Sciences, 2020). This point should be discussed.
  15. Epigenetic mechanisms such as DNA methylation, histone modifications, and microRNAs impact an array of transcriptional responses and, at least in part, represent a potential mechanism that might explain microglial sex-related functional differences (Uncovering sex differences of rodent microglia, Journal of Neuroinflammation, 2021). This point should be discussed.
  16. The authors linked changes in microglial activation and phenotype with changes in the metabolic profile of cells. But how about functional changes of these microglia (cytokine production and phagocytosis) in male and female after DMF treatment? This is major concern.
  17. ECAR was decreased in microglia prepared from DMF-treated, compared with vehicle-treated, female mice, whereas there was an increase in ECAR in microglia prepared from DMF-treated male mice compared with vehicle-treated male mice. Primary microglia culture (with and without sex hormones) should be used to confirm these findings.
  18. How many times were repeated in each experiment? This should be mentioned in the figure legends.
  19. Would these sex-specific effects of DMF be noted in human?

Round 2

Reviewer 1 Report

Check for typos, there are several mistakes.

Reviewer 3 Report

The authors have addressed my concerns.